# Residue Depletion of Imidocarb in Bovine Tissues by UPLC-MS/MS

**DOI:** 10.3390/ani13010104

**Published:** 2022-12-27

**Authors:** Yaoxin Tang, Na Yu, Chunshuang Liu, Mingyue Han, Honglei Wang, Xiaojie Chen, Jijun Kang, Xiubo Li, Yiming Liu

**Affiliations:** 1National Feed Drug Reference Laboratories, Feed Research Institute, Chinese Academy of Agriculture Sciences, Beijing 100081, China; 2Laboratory of Quality & Safety Risk Assessment for Products on Feed-Origin Risk Factor Ministry of Agriculture and Rural Affairs, Feed Research Institute, Chinese Academy of Agriculture Sciences, Beijing 100081, China

**Keywords:** protozoal, bovine, imidocarb, residue depletion, UPLC-MS/MS, withdrawal time

## Abstract

**Simple Summary:**

Bovine protozoal disease is a common disease that causes huge economic losses to the cattle breeding industry. Imidocarb (IMD) can be used to treat protozoal parasite diseases in cattle including babesiosis, anaplasmosis, and eperythrozoonosis. However, residues of imidocarb in bovine tissues pose a risk to food safety. This study conducted a residue depletion of imidocarb in cattle based on an ultra-high performance liquid chromatography-tandem mass spectrometry method and obtained the IMD withdrawal time in cattle by the calculation through the WT1.4 software. The results show that the liver and kidney may be the target tissues for IMD and the withdrawal time of IMD in cattle is 224 days at the dosage of 3.0 mg/kg. This will guide farmers in the proper use of imidocarb and ensure human food safety.

**Abstract:**

In this study, an ultra-high performance liquid chromatography-tandem mass spectrometry (UPLC-MS/MS) method was developed for the residue depletion of imidocarb (IMD) in bovine tissues, and the drug withdrawal time of IMD was determined. Twenty-five clinically healthy cattle (body weight 300 kg ± 15 kg) were randomly divided into five groups of five cattle each. The cattle were treated subcutaneously injecting a single dose of a generic IMD formulation, at the recommended dosage of 3.0 mg/kg. The five groups of cattle were slaughtered respectively at 96, 160, 198, 213, and 228 days after IMD administration. Samples from the liver, kidney, muscle, fat, and injection site were collected from each animal. After subtilis proteinase was used to digest the tissue, the content of IMD in the samples was analyzed by UPLC-MS/MS method. In conclusion, the method validation results showed that the method meets the criteria, and the longest withdrawal time of 224 days for the liver can be selected as the conclusive withdrawal time to guarantee consumer safety.

## 1. Introduction

Bovine parasitic disease is a common disease and causes huge economic losses to the cattle breeding industry. Bovine protozoa mainly parasitize bovine cells and cause clinical symptoms in cattle. Babesia parasites, anaplasma marginale, and eperythrozoonosis are serious diseases caused by bovine protozoa [1,2]. Babesia parasites and anaplasma are prevalent tick-borne pathogens in cattle that infect erythrocytes in the hosts. Both induce acute and persistent infections, cause hemolytic diseases in cattle, and even lead to death in the most severe cases [3,4].

Imidocarb (IMD) is a carbanilide derivative with molecular formula C_19_H_20_N_6_O and the molecular structure is shown in Figure 1. The dipropionate salt of imidocarb (IMDP) can be used to treat protozoan parasite diseases including babesiosis, anaplasmosis and eperythrozoonosis [5,6,7].

Research on veterinary drug residues in edible animal tissues is of fundamental importance for public health. The maximum residue limit (MRL) is an important index to measure the food safety of animal products. Codex Alimentarius Commission (CAC) established the MRLs for IMD in bovine edible tissues as follows: 50 μg/kg in fat, 300 μg/kg in muscle, 2000 μg/kg in muscle, and 1500 μg/kg in the liver (2005). It has been recently reported that IMD exhibits a favorable kinetic profile in cattle. When subcutaneously administered at 3.0 mg/kg, IMD is rapidly absorbed in the blood and reaches a plateau in about 2 h. IMD elimination from plasma is slow ($T_{1/2}$ = 31.77 ± 25.75 h), and the apparent volume of distribution in cattle was 6.53 ± 5.34 mL/g [8]. The slow elimination of IMD in bovine tissues is conducive to protozoa elimination, but it poses risks and challenges to food safety.

There have been several studies on the residual depletion of IMD in bovine tissues using different determination methods. Coldham et al. have set up and validated a high-performance liquid chromatography (HPLC) based method to investigate the retention of IMD in the bovine liver, and have subsequently studied the half-life of IMD in the bovine liver and muscle [9,10]. Tarbin et al. have developed an HPLC method for the determination of the antiprotozoal agent IMD in bovine kidneys [11]. Park et al. have used high-performance liquid chromatography (HPLC) with diode-array detection (DAD) to study residue depletion of IMD in beef and milk [12]. Traynor et al. have determined IMD in bovine tissues and milk samples by LC-MS/MS [5]. However, there has yet to be a UPLC-MS/MS to study IMD residual depletion in the bovine liver, kidney, muscle, fat, and injection sites.

The aim of this study was to obtain residual parameters in different bovine tissues (liver, kidney, muscle, fat, and injection site) with optimized UPLC-MS/MS conditions and to determine the conclusive drug withdrawal period to ensure food safety as set in CAC’s MRLs regulations.

## 2. Materials and Methods

### 2.1. Animals

Twenty-five clinically healthy Luxi male cattle, aged almost 6 months and weighing 300 ± 15 kg, were randomly divided into five groups of five cattle each before IMD administration. The cattle were routinely fed for half a month with materials containing no antibacterial drugs, insect resistance, and daily drinking water. The cattle were treated subcutaneously by a single dose injection with a generic IMDP formulation, at the recommended dosage of 3.0 mg/kg [9]. All animal experiment procedures were approved by the Animals Use and Care Committee of Feed Research Institute, Chinese Academy of Agricultural Sciences (number: FRI-CAAS-20150527) and performed in conformity with its regulations.

### 2.2. Sample Collection

The cattle treated with IMDP were slaughtered in groups of five animals at 96, 160, 198, 213, and 228 days after drug administration [8]. Separate samples from muscle, injection site, fat, liver, and kidney were obtained. The samples were labeled and stored at −20 °C until analysis.

### 2.3. Sample Preparation

Samples (liver, kidney, muscle, injection site) of 1.0 ± 0.02 g were weighed and placed in a 50 mL polypropylene centrifuge tube, adding 1 mL PBS buffer solution, and vortexing for 1 min. A total of 300 μL subtilis protease solution was added and vortexed for 2 min, and the samples were oscillated at 37 °C for 6 h. A total of 2 mL PBS buffer solution was added, then the samples were vortexed for 1 min, ultrasonicated for 15 min, and centrifuged for 10 min at 7500 g. The residue was re-extracted twice, and the combined extracted solution was passed through the SPE column (Waters Oasis WCX, 3cc 60 mg, Waters Company, Milford, MA, USA) previously activated with 3 mL methanol and balanced with 3 mL water. The SPE column was washed with 3 mL cleaning solution and was eluted with 3 mL methanol/formic acid (96/4, *v*/*v*). The eluent was collected into 10 mL glass tubes. The eluent was concentrated to dryness in a water bath at 40 °C with nitrogen. The residue was redissolved with 1 mL of methanol/water (15/85, *v*/*v*) and vortexed for 1 min, then filtered through a 0.22 μm filter membrane, the filtrate was placed in the injection vial until analysis. For fat samples, which could not be decomposed well by the Bacillus subtilis proteinase, 300 μL extract (0.2% EDTA solution/methanol/acetonitrile, 1/8/1, *v*/*v*/*v*) was used to deal with the fat samples. The other steps are the same as above.

### 2.4. Reagents

IMD injection (85 mg/100 mL) was provided by Qilu Animal Health Products Corp. Ltd. (Jinan, China). The reference drug (the marketed IMD, 98.5%) was manufactured and provided by J&K Scientific Corp. LTD (Beijing, China). Acetonitrile (ACN) and methanol from Fisher Scientific (Fair Lawn, NJ, USA) were liquid chromatography/mass spectrometer (LC/MS)-grade. Formic acid (HPLC) was purchased from ERA (Austin, TX, USA). Trifluoroacetic acid (AR) and ammonia (AR) were obtained from Sinopharm Chemical Reagent Co., Ltd (Shanghai, China). The SPE column was purchased from WATERS Oasis (USA). Subtilisin Carlsberg was from Sigma (Poole, Dorset, UK). A cylinder-type microporous filter membrane (WAT200810, 0.22 μm) was obtained from WATERS Oasis (USA). Ultrapure water was prepared by PALL Cascada II I (Show Low, AZ, USA).

### 2.5. UPLC-MS/MS Conditions

A Waters ACQUITY UPLC and Xevo TQ-S Mass Detector (Waters company, Milford, MA, USA) was used as UPLC-MS/MS system. Chromatographic separation was achieved on a C18 reverse-phase column (ACQUITY UPLC^®^ BEH Phenyl column 1.7 μm, 50 mm × 2.1 mm). The column temperature was set at 40 °C. Gradient elution was established with a mobile phase consisting of methanol (eluent A) and 0.1% (*v*/*v*) formic acid in water (eluent B). The injection volume was 2 μL, and the flow rate was 0.35 mL/min. At the start of the run, A ran at 15% for 1.5 min, increasing linearly from 15 to 90% for 1.5 to 3 min and then reduced linearly from 90 to 15% for 3 to 5 min. A ran at 15% for one minute, a total run time of 6 min. MS analyses were carried out using multiple reaction monitoring (MRM) with an electrospray ionization (ESI) source in the positive mode. The MS parameters are provided in Table 1, and the MRM parameters are provided in Table 2. MassLynx version 4.0 software running in Microsoft Windows 7 Professional was used to operate the instrument for data acquisition and data processing for automated quantification.

### 2.6. Method Validation

Method validation was performed based on VICH GL49. The following parameters were assessed: selectivity, linearity, precision, accuracy, limit of detection (LOD), limit of quantification (LOQ), and stability.

#### 2.6.1. Selectivity and Matrix Effect

Selectivity is the ability of a method to distinguish the analyte being measured and other substances that might be present in the sample. In this study, selectivity was evaluated by comparing the chromatograms of IMD-free tissue samples with those of corresponding tissue spiked with IMD. Matrix effects were assessed by comparing the peak area response of treated blank tissue samples spiked with IMD with an equivalent concentration of IMD standard solution directly dried and reconstituted in the same mobile phase.

#### 2.6.2. Linearity

The IMD standards were added into control matrix extracts (liver, kidney, muscle) to prepare a series of matrix-matched standard solutions with concentrations of 1, 5, 10, 20, 25, and 50 ng/mL. The IMD standards were added to control matrix extract (fat) to prepare a series of matrix-matched standard solutions with concentrations of 1, 10, 25, 50, and 75 ng/mL. The calibration curves were plotted with the peak area of IMD as the y-axis and the nominal concentration of IMD as the *x*-axis. The measured drug peak area (Y) and drug concentration in tissue samples (X) were used for linear regression analysis by weighted $\left(1/x^{2}\right)$ least squares method.

#### 2.6.3. LOQ and LOD

The limit of detection (LOD) and the limit of quantifications (LOQ) were defined by adding a known concentration of IMD into the IMD-free tissue samples, and the lowest concentrations met the requirements of signal-to-noise ratios (S/N) of ≥3 and ≥10 [13], respectively.

#### 2.6.4. Precision and Accuracy

The precision was expressed as the relative standard deviation (RSD) or the coefficient of variation (CV). The accuracy, expressed as analyte recovery, indicates the closeness between the measured concentration and the true value of the analyte concentration. Intra-day and inter-day precision were calculated by analyzing quality control (QC) concentrations (1/2MRL, MRL, 2MRL of each tissue) in five replicates per concentration on the same day and five measurements of each concentration conducted over three days.

#### 2.6.5. Stability

Examining the stability of drugs added to tissue samples mainly includes the examination of long-term stability, short-term stability, and repeated freeze–thaw stability. Long-term stability was investigated by adding blank tissue homogenate samples to the compounds to be tested, vortexing thoroughly and storing them in a −20 °C refrigerator, and taking out the mixture after 15 days for natural thawing. Short-term stability was investigated by taking out the mixture after 7 days of freezing and thawing, and repeated freeze–thaw stability was investigated by taking out the mixture after 24 h of freezing and thawing, and storing them in a −20 °C refrigerator after thawing, and repeating the procedure three times. In this study, two different concentrations (muscle, liver, kidney: 2 and 50 μg/kg, fat: 2 and 100 μg/kg) were added for each tissue, and three parallel samples were set up at one concentration.

## 3. Results

### 3.1. Selectivity and Matrix Effect

No interference was observed in the retention time corresponding to IMD (1.1 min) in any of the bovine tissues (muscle, liver, kidney, and fat) analyzed. This confirmed the selectivity of our analytical method (Figure 2). To assess the effect of the bovine tissue matrix on IMD, chromatograms of 10 ng/mL IMD standard solution were compared with chromatograms of blank bovine tissue samples (muscle, liver, and kidney) spiked with 10 ng/mL IMD standard solution, and bovine fat samples spiked with 25 ng/mL IMD standard solution, as shown in Figure 2. The results showed that the bovine tissue matrix could increase the IMD peak area.

### 3.2. Linearity

The results of the matrix effect showed that the bovine tissue matrix increases the IMD peak area; therefore, we added a control tissue matrix to eliminate the matrix effect in this study. The calibration curve for IMD was linear over the concentration range of 1–50 ng/mL according to the results of a weighted ($1/x^{2}$) least-square linear regression analysis (Figure 3). The linear correlation coefficient values (r) were all >0.99 (Table 3), indicating a strong correlation between the IMD concentration level and the analytical response. The calibration equations for bovine tissues are shown in Table 3, where *Y* is the peak area and *X* is the concentration of IMD in ng/mL.

### 3.3. LOD, LOQ, and Stability

The LODs in the liver, kidney, and muscle were all 0.5 μg/L according to S/N ≥3. The LOQs in the liver, kidney, and muscle were all 1 μg/L according to S/N ≥10. The results for long-term stability, short-term stability, and repeated freeze–thaw stability are shown in Table 4. The results showed that IMD in four different bovine tissues was stable after storage at −20 °C for up to 15 days and after repeated freeze–thaw cycles for 3 days. The mean values met the criteria for residue testing.

### 3.4. Accuracy and Precision

The recoveries of IMD’s mass spectrometry in each tissue and the coefficients of variation (CV) for the inter-day and intra-day are shown in Table 5. The recoveries of three concentrations in liver tissue ranged from 78.1 to 89.6%. The intra-day coefficient of variation was 2.5–4.8%, and the inter-day coefficient of variation was 3.0–3.9%. The recoveries of three concentrations in kidney tissue ranged from 77.8 to 88.6%. The intra-day coefficient of variation was between 0.5% and 5.6%, and the inter-day coefficient of variation was between 2.9% and 4.7%. The recoveries of three concentrations in muscle tissue ranged from 79.7 to 80.7%. The intra-day coefficient of variation was 1.6–4.9%, and the inter-day coefficient of variation was 2.6–5.1%. The recoveries of three concentrations in fat tissue ranged from 84.3 to 88.5%. The intra-day coefficient of variation was 2.7–15.0%, and the inter-day coefficient of variation was 6.6–13.4%. The results showed that the method was accurate and precise, and could satisfy the determination of IMD in bovine tissues.

### 3.5. Residue Concentration of IMD in Bovine Tissue Samples

The bovine tissue samples were processed according to the above methods and then detected by UPLC-MS/MS method as described. Mean tissue concentration data in cattle in the different experimental groups are shown in Table 6.

### 3.6. Withdrawal Period of IMD

The withdrawal time of IMD in each tissue was calculated using the WT 1.4 software, referring to the “Guideline on determination of withdrawal periods for edible tissues” developed by EMEA (2018). The depletion curve (Figure 4) was prepared with the average residue concentrations in the bovine liver, kidney, muscle, fat, and injection sites and the withdrawal time. The withdrawal times for bovine tissues were 223.23 days for the liver, 166.86 days for the kidney, 151.78 days for muscle, 151.78 days for fat, and 156.78 days for the injection site. The time points do not make up a full day, so the withdrawal periods have to be rounded up to the next day. Therefore, the conclusive withdrawal times for bovine tissues were 224 days for the liver, 167 days for the kidney, 152 days for muscle and fat, and 157 days for the injection sites.

## 4. Discussion

Protozoan parasites, such as Babesia parasites, Anaplasma marginale, Plasmodium, and Toxoplasma, have been posing a threat to human and animal health for a long time [14]. Bovine babesiosis, caused by Babesia bovis or Babesia bigemina, is a disease characterized by significant cattle morbidity and mortality, resulting in considerable economic losses to the livestock industry. Infection in young cattle is usually asymptomatic, but it leads to hemoglobinemia, hemoglobinuria, anorexia, fever, abortion, and death in adult cattle [15]. Bovine anaplasmosis, caused by Anaplasma marginale, is an intraerythrocytic rickettsia and it is characterized by mild to severe hemolysis and anemia that affects animal health and reproduction [4,16]. Currently, there are only a few drugs available on the market to treat parasitic diseases, and alternative treatment options are limited when parasites are resistant to drugs [17]. Imidocarb, administered as dipropionate salt, has been widely used in animal husbandry and has been recognized as an effective drug against protozoan parasites in the international community so far. A few studies have suggested some species of animals may have adverse reactions to IMD. IMD may cause salivation, vomiting, diarrhea, and dyspnea in animals, and especially in equines, it may cause neurotoxic and allergic reactions [18,19]. It has been reported that a dog died of bronchoconstriction secondary to acetylcholinesterase inhibition after intravenous treatment with IMD [20]. IMD inhibits the activity of acetylcholinesterase (AChE) and adenosine deaminase (ADA) after subcutaneous injections in cattle [21]. In addition, the high and persistent IMD concentrations observed in sheep brains indicate that the drug is able to cross the blood–brain barrier, which raises concerns about the potential neurotoxic effects of IMD [22].

Drug residues in foods of animal origin are considered a public health risk. The FAO (Food and Agriculture Organization) and WHO (World Health Organization) report that in developing countries the average amount of drug residues in edible animal tissues exceeds permissible levels. Beef containing drug residues may pose mild to severe risks to humans, having the potential to cause allergic reactions and may have mutagenic, teratogenic, and carcinogenic effects [23,24]. To this day, there are no data about adverse reactions caused by IMD ingestion in humans; however, based on the above adverse reactions of IMD in different animals, and especially considering the inhibitory effect on acetylcholinesterase and the ability to cross the blood–brain barrier, it is advisable to carefully research the presence of IMD residues in animal tissues intended for nutrition.

In this study, a new UPLC-MS/MS method was established. According to literature reports, for the extraction methods of IMD from bovine tissues, the liquid–liquid extraction method and solid-phase extraction method using acetonitrile or acetone as the extraction solution are the most commonly adopted to extract IMD from bovine tissues [11,25,26]. However, in this study, when acetonitrile (0.1% trifluoroacetic acid), 80% acetonitrile (0.1% trifluoroacetic acid), and 30% trifluoroacetic acid solution were used as extraction solution, the solid-phase extraction method was used to extract IMD in bovine tissues, and the recovery rates were found to be less than 60%. Coldham et al. have adopted subtilisin Carlsberg to digest tissues and subsequently used acetonitrile to extract the drug from the tissues. The recovery rate observed was about 70% [9]. Therefore, the enzymatic hydrolysis method was used to compare the extraction efficiency of β-Glucosidase and Bacillus subtilis proteinase and it was found that the latter had a higher recovery rate and fewer impurities. In addition, after the tissues were digested with proteases, there were no significant differences in the recovery rates between the extraction solution passed through the WCX column after re-extracted with methanol and acetonitrile and the extraction solution passed directly through the column. We, therefore, decided to use the method of extraction solution directly passed through the column after subtilis protease digestion of the tissue. The recovery rate was low in fat tissues when the enzymatic hydrolysis method was used, probably because of the poor lipolysis obtained by subtilis proteinase. However, when 0.2% EDTA:methanol:acetonitrile (1:8:1) was used as the extraction solution, the recovery rate of IMD in fat was the highest.

According to the ${\left[M+H\right]}^{+}$ abundance of the drug molecule in ESI+ ionization mode, the ion (m/z 349.1) was regarded as the precursor ion. After optimizing the mass spectrum parameters of the precursor ion (m/z 349.1), the secondary scanning mode was used to find the product ions. Three major fragment ions were found (m/z 187.98, m/z 161.96, and m/z 94), among which fragment ion (m/z 187.98) had the largest abundance and was selected as the quantitative ion. The fragment ion (m/z 161.96) with the second highest abundance was a qualitative ion. However, in our study, we found that when the ion (m/z 349.1) was used as the precursor ion, the intensity of the matrix effect was high, which seriously interfered with the detection of the sample. IMD is easily ionized in aqueous solution, and there are two main ionized forms (monoprotonated m/z 349.1 ${\left[M+H\right]}^{+}$; deprotonated m/z 175 ${\left[M+2H\right]}^{2+}$) [27]. In practice, the abundance of IMD was stronger and more stable when m/z 175 was selected as the precursor ion. Therefore, the deprotonated m/z 175 ${\left[M+2H\right]}^{2+}$ was set to the precursor ion in our study.

The validation parameters obtained for the analytical procedures selected to detect IMD in the different tissues are in agreement with VICH GL49(R) (2015) recommended by the Food and Drug Administration, ensuring that these procedures are suitable for the intended purpose. The linearity, accuracy, precision, selectivity, specificity, LOQ, and LOD of the present study all meet the criteria. After a single subcutaneous injection of IMD at 3 mg/kg body weight, IMD residue in the liver was the highest, followed by the kidneys. These results suggest that the liver and kidney may be the target tissues for IMD residues, which is consistent with the data reported for swine, horses, and sheep [13,28,29]. At 228 d, the mean residual concentrations of IMD in the liver and kidney were 249.6 μg/kg and 131.6 μg/kg. Coldham had previously administered the same dose to cattle and IMD was detectable in the liver on 224 days, which is consistent with our results [9]. The long residual time of IMD in the liver, kidney, and other edible tissues of cattle may be due to its strong combination with DNA in cells and its slow biotransformation process [9,30]. In 2005, Codex Alimentarius Commission (CAC) set the maximum residue limits (MRLs) for IMD in bovine tissues at 2000 μg/kg in the kidney, 1500 μg/kg in the liver, 300 μg/kg in muscle, and 50 μg/kg in fat. At the first sampling point, 96 days after IMD administration, the drug concentration in the kidney was lower than MRL in four of the five cattle. At the third sampling point, 198 days after administration, the IMD concentrations in the liver of all five animals were lower than MRL. Drug concentrations in muscle, fat, and injection sites were below MRL 160 days after administration. The results calculated with the WT1.4 software showed a withdrawal time of 224 days for the liver and 167 days for the kidney, which was much higher than the withdrawal time in the swine liver (54 days) and kidney (34 days). In order to guarantee consumer safety, the longest withdrawal time of 244 days in the liver can be selected as the conclusive withdrawal time in bovine tissues.

## 5. Conclusions

A processing method and UPLC-MS/MS method were established for the detection of IMD in bovine tissues. The results of the method validation showed that the selectivity, linearity, LOD, LOQ, precision, accuracy, and stability of the method all meet the criteria. The results showed that the highest IMD residues were found in the liver, followed by the kidney, suggesting that the liver and kidney may be the target tissues of IMD. Based on the residual depletion of IMD in different tissues, the withdrawal time after IMD treatment in cattle was calculated to be 224 days. These results provide guidance for the rational use of IMD and ensure food safety.

## Figures and Tables

**Figure 1 animals-13-00104-f001:**
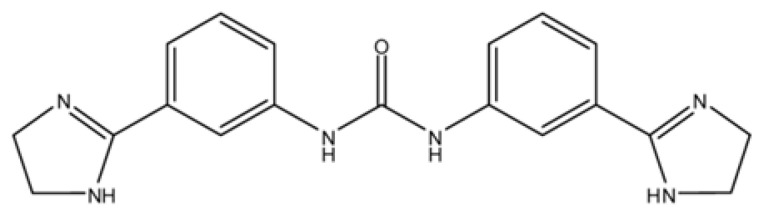
Molecular structure of IMD.

**Figure 2 animals-13-00104-f002:**
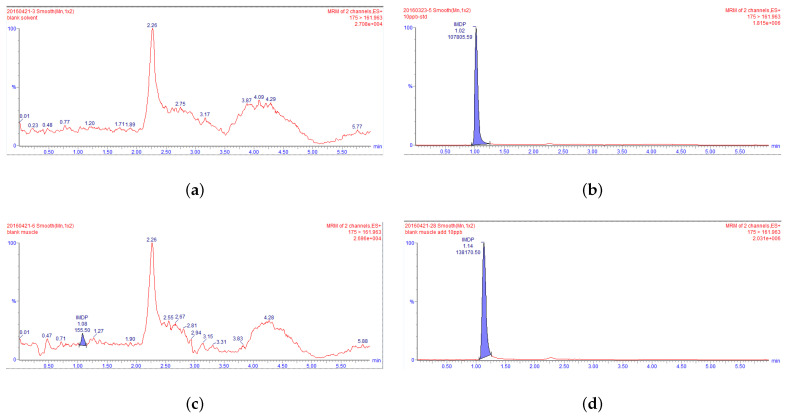
UPLC-MS/MS chromatograms. (**a**) for a blank mobile phase, (**b**) for a blank mobile phase spiked with IMD (10 ng/mL), (**c**) for a blank muscle tissue sample, (**d**) for a blank muscle tissue sample with IMD (10 ng/mL), (**e**) for a blank liver tissue sample, (**f**) for a blank liver tissue sample spiked with IMD (10 ng/mL), (**g**) for a blank kidney tissue sample, (**h**) for a blank kidney tissue sample spiked with IMD (10 ng/mL), (**i**) for a blank fat tissue sample, and (**j**) for blank fat tissue sample spiked with IMD (25 ng/mL).

**Figure 3 animals-13-00104-f003:**
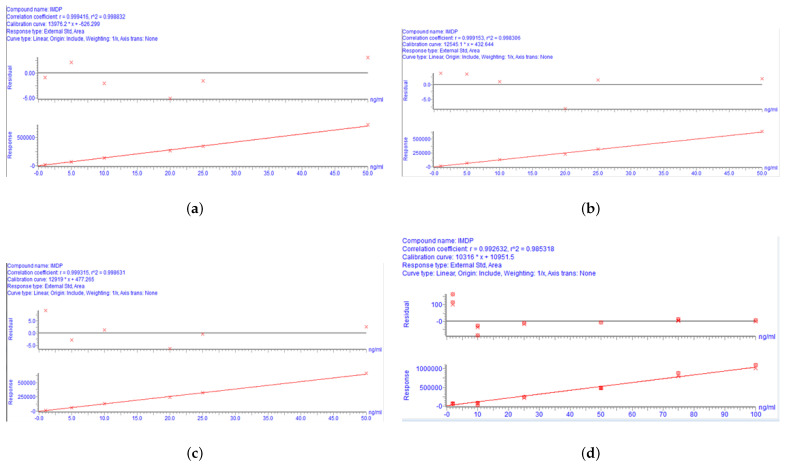
The calibration curves for IMD in bovine tissues: (**a**) for muscle, (**b**) for the kidney, (**c**) for the liver, and (**d**) for fat.

**Figure 4 animals-13-00104-f004:**
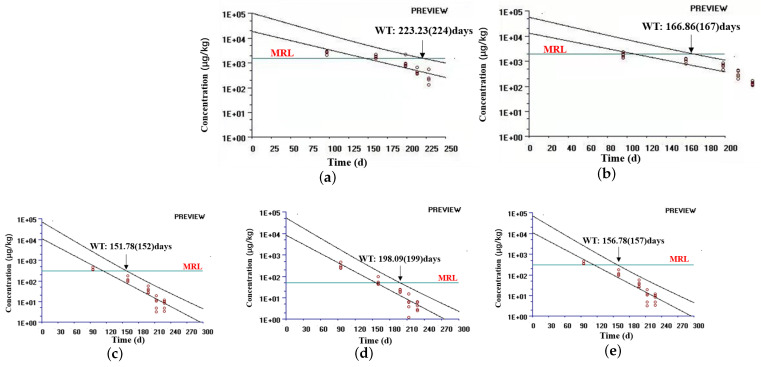
Estimated withdrawal times for IMD in bovine tissues: (**a**) for the liver, (**b**) for the kidney, (**c**) for muscle, (**d**) for fat, and (**e**) for the injection site.

**Table 1 animals-13-00104-t001:** MS parameters.

Parameter	Settings
Ionization mode	Electrospray ionization (positive mode)
Capillary voltage	2.92 kV
Desolvation temperature	500 °C
Cone gas flow	150L/h
Desolvation gas flow	850L/h
Secondary collision gas	Ar_2

**Table 2 animals-13-00104-t002:** MRM parameters.

Precursorion (*m*/*z*)	Production (*m*/*z*)	Cone Voltage (V)	Collision Energy (eV)
175	161.96 ^1^	30	16
175	187.98 ^2^	30	16

^1^ Qualitative identification ion. ^2^ Quantitation ion.

**Table 3 animals-13-00104-t003:** The calibration equation and correlation coefficients for IMD in tissues.

Tissue	Calibration Equation	r
Muscle	y = 13976.2x − 626.299	0.9994
Kidney	y = 1545.1x + 432.644	0.9991
Liver	y = 12919x + 477.265	0.9993
Fat	y = 10316x + 10951.5	0.9926

**Table 4 animals-13-00104-t004:** Stability results of IMD in four tissues of cattle.

Tissue	Concentration (μg/kg)	Measured Concentration (μg/kg) of IMD (Mean ± SD, n = 3)			
		0 d	7 d (−20 °C)	15 d (−20 °C)	Repeated Freeze–Thaw
Muscle	2	1.70 ± 0.08	1.67 ± 0.05	1.63 ± 0.12	1.60 ± 0.14
	50	44.60 ± 3.02	44.53 ± 3.97	44.33 ± 1.30	45.33 ± 1.67
Fat	2	1.77 ± 0.19	1.60 ± 0.08	1.70 ± 0.22	1.60 ± 0.08
	50	83.57 ± 6.87	86.23 ± 3.93	93.57 ± 0.39	88.07 ± 11.10
Liver	2	1.90 ± 0.14	1.73 ± 0.12	1.80 ± 0.08	1.60 ± 0.16
	50	43.47 ± 4.29	44.80 ± 3.40	44.50 ± 5.26	45.03 ± 0.71
Kidney	2	1.67 ± 0.05	1.47 ± 0.05	1.87 ± 0.05	1.73 ± 0.09
	50	44.27 ± 1.30	44.70 ± 3.61	46.90 ± 0.67	45.93 ± 3.70

**Table 5 animals-13-00104-t005:** The average recovery, intra RSD, and inter RSD of IMD in cattle tissues at 3 spiked concentrations (μg/kg).

Tissue	Spiked Concentration	Average Recovery (%)	SD (%)	Intra RSD (%)			Inter RSD (%)
				1 d	2 d	3 d	
Muscle	150	78.1	1.4	3.3	3.3	5.2	3.9
	300	80.1	0.6	4.6	4.4	2.8	3.5
	600	89.6	0.2	2.8	2.5	4.8	3.0
Kidney	1000	77.8	1.4	2.3	3.8	2.6	3.1
	2000	84.1	1.5	4.8	4.9	5.6	4.7
	4000	88.6	2.0	0.5	2.9	2.2	2.9
Liver	750	80.7	2.9	2.3	4.6	4.9	5.1
	1500	79.7	0.9	2.9	4.5	3.1	3.2
	3000	80.7	0.9	3.8	2.5	1.6	2.6
Fat	25	84.3	0.2	10.2	6.4	2.7	7.7
	50	88.5	3.1	5.5	2.8	9.6	6.6
	100	86.3	4.3	5.6	15.0	10.2	13.4

**Table 6 animals-13-00104-t006:** IMD concentrations (μg/kg) in bovine tissue samples.

Days-Post Treatment	Liver ± SD	Kidney ± SD	Muscle ± SD	Fat ± SD	Injector Site ± SD
96	2644.00 ± 316.91	1804.00 ± 369.16	396.00 ± 72.18	313.70 ± 95.74	436.00 ± 94.29
160	1840.00 ± 249.70	1106.00 ± 217.78	124.80 ± 22.77	57.93 ± 21.13	117.00 ± 34.26
198	1116.00 ± 602.10	650.40 ± 147.53	21.32 ± 15.27	31.60 ± 13.43	32.52 ± 5.77
213	497.00 ± 148.94	316.00 ± 101.66	9.60 ± 4.96	6.54 ± 5.26	10.06 ± 6.49
228	249.60 ± 167.94	131.60 ± 23.64	4.40 ± 1.71	4.50 ± 1.94	7.62 ± 3.54

## Data Availability

The data presented in this study are available on request from the corresponding author.

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
