# Peer review of "Residue Depletion of Imidocarb in Bovine Tissues by UPLC-MS/MS"

_animals, 2022, doi:10.3390/ani13010104_

Round 1
Reviewer 1 Report
This work by Tang et al describes the study of residue depletion of imidocarb in cattle based on an ultra-high performance liquid chromatography-tandem mass spectrometry method, and obtained the IMD withdrawal time in cattle by the calculation through WT1.4 software.
Although the work seems interesting, it needs to undergo extensive changes, as follows:
1. English needs to be checked and rectified.
2. Figures are of poor quality (especially Fig. 2). Replace them with high-resolution images.
3. Conclusion can be more elaborative.
This can be accepted after addressing above comments.
Author Response
Thank you for your advice. All your suggestions are very important.
Response 1: We have checked the full language and corrected the incorrect statements. For details, see the reuploaded article.
Response 2: High-resolution images have been re-provided.
Response 3: The conclusion was rewritten and sublimated.
Thank you again for your advice.
Reviewer 2 Report
Residue depletion of imidocarb in bovine tissues by UPLC-MS/MS
The article's subject is relevant to human food safety. The text approaches the subject with scientific correctness. The article is well organized, and it is simple to follow. Despite this, some improvements can be made to make the text clearer. The Tables and Figures are relevant for understanding the article. The material and methods are clearly described, which will allow for a perfect understanding by other researchers. The results are well discussed with the existing knowledge on the subject. Finally, the results support the conclusions.
Some detailed comments are below:
L18-19. In conclusion, the results of the method validation showed that the method meet the criteria and the longest withdrawal time of 224 days for liver can be selected as the conclusive withdrawal time to guarantee consumer safety. “please change with” In conclusion, the method validation results showed that the method meets the criteria, and the longest withdrawal time of 224 days for the liver can be selected as the conclusive withdrawal time to guarantee consumer safety.
L27 infections and cause hemolytic “please change with” infections, cause hemolytic
L38 1500µg/kg in liver “please change with” 1500µg/kg in the liver
L42 .The slow . The slow. Please check for similar issues and add a space. L176 Table 3,where….. L290 .The long
L52 However, there is no report “please change with” However, there has yet to be a
L61-62 The cattle were routinely fed for half a month with the materials did not contain any antibacterial drugs and insect resistant and the daily drinking water. “please change with” The cattle were routinely fed for half a month with materials containing no antibacterial drugs, insect resistance, and daily drinking water.
L67 conformity to its regulations “please change with” conformity with its regulations
L148 The examination of the stability “please change with” Examining the stability
L171 area, therefore, we “please change with” area; therefore, we
L181 stability were shown “please change with” stability are shown
L208 Withdrawal time “please change with” The withdrawal time
L212 tissues were: 224 days for liver,167 “please change with” tissues was: 224 days for liver, 167
L223 Currently there “please change with” Currently, there
L235 about potential neurotoxic “please change with” about the potential neurotoxic
L257 rate and less impurities “please change with” rate and fewer impurities
L261 We therefore “please change with” We, therefore,
L264 lipolysis obtained by “please change with” lipolysis was obtained by
L287 horses, sheep[“please change with” horses, and sheep [
Author Response
Thank you for your advice. All your suggestions are very important.
All your comments are of great help to my writing.
We have made careful revision according to all your comments.
Reviewer 3 Report
Journal: Animals
Manuscript ID: animals-2054595
Title: Residue depletion of imidocarb in bovine tissues by UPLC-MS/MS
Comments to the Author
In the study, the authors deal with the residue of imidocarb in bovine tissues including, liver, kidney, muscle, fat and injection site after subcutaneous injection of imidocarb at a dosage of 3 mg/kg body weight in cattle. Overall, the manuscript is well written. The experimental design and analysis have no major flaws. The results from this study are informative for veterinary practice and food safety. Therefore, I am thinking that it can be acceptable for publication after revision since I have some points in the manuscript should be clarified.
1.) Animal age, gender and bleed should be clarified.
2.) The dosage of 3 mg/kg body weight was used for subcutaneous injection, the reference is needed.
3.) Why were tissues collected at 96, 160, 198, 213 and 228 days after drug administration?
4.) An internal standard was not used in the assay. Some reviewers insist on an internal standard, although this is not always necessary in modern LC-MS. Explain why this may not have been needed here to make sure the results of analytical method is reliable.
5.) I am thinking that the matrix-matched calibration curve does not match to the results in table 6. This reviewer would like to know how to quantitate the level of drug in the sample when it was over 50 ng/ml.
6.) Together with this, it would be interesting, if the authors can show the %SSE or %ME in the manuscript. These do not clear for this reviewer.
Author Response
Thank you for your advice. All your suggestions are very important.
Response 1: "Twenty-five clinically healthy Luxi male cattle, aged almost 6 months and weighing 180±15 kg, were randomly divided into five groups of five cattle each before IMD administration."
Response 2: Relevant references have been cited.
Response 3: The IMD injection (AKZO-NOBEL Corp.) specification states that the withdrawal time of the drug is 213 days. Studies have shown that the IMD injection (Qilu Animal Health Products Corp. LTD) is bioequivalent to the IMD injection (AKZO-NOBEL Corp.). On the basis of 213 days, ±15 days are 198 days and 228 days, 96 days is about 3 months after administration, 160 days is about 5 months after administration, and 213 days is about 7 months after administration, so these five time points are selected.(References have been cited in this article.)
Response 4: During our experiment, no suitable internal standard was found. And because the machine is automatic sampling, the external standard method can meet the experimental requirements.
Pesponse 5: The concentration of IMD standard can meet the results in Table 6, but it is diluted when making the curve.
Thank you again for your advice.